# Non-linear impacts of local fiscal expenditure on farmers' income: A SHAP-informed machine-learning analysis

**Tingting Zhang** [ID], **Jinshuai Zhang** [ID]*

School of Accountancy, Anyang Institute of Technology, Anyang, China

* 20242008@ayit.edu.cn

## Abstract

Understanding how local fiscal spending shapes rural income is central to China's rural revitalisation strategy. Using panel data from 17 prefecture-level cities in Henan Province for 2010–2023, this study investigates the non-linear and heterogeneous effects of fiscal expenditure on farmers' per-capita income. A hybrid econometric–machine learning framework is developed, combining city–year fixed effects with a residual XGBoost learner and SHapley Additive exPlanations (SHAP) to capture both the linear baseline relationships and complex non-linear interactions among fiscal items while retaining interpretability. Model evaluation based on repeated nested cross-validation and 500 permutation tests demonstrates high predictive reliability (out-of-sample $R^2 = 0.924$, $p = 0.002$). SHAP-based analysis reveals that health-care and education spending are the dominant determinants of rural income, jointly accounting for over 40% of model influence. Partial-dependence plots uncover clear threshold effects: education and health expenditures exhibit inverted-U shapes with turning points at approximately ¥1,800 and ¥1,050 per rural resident (2015 prices), respectively. Infrastructure investment shows consistently positive but diminishing returns, while social-security transfers produce concave yet non-negative effects. Heterogeneity analysis further indicates that low-capacity cities derive greater benefits from technology and transport spending, whereas high-capacity cities gain more from health and urbanisation budgets. Robustness tests across alternative learners (Random Forest, LightGBM) and variable definitions confirm the stability of these non-linear thresholds. The results highlight the importance of optimising fiscal composition rather than merely increasing total spending, suggesting that city-specific expenditure ceilings—particularly for education and health—could raise rural incomes by 2–3% while enhancing fiscal efficiency.

**Data availability statement:** Replication data and code can be found in GitHub: https://github.com/ZJS0201/The-impact-of-public-financial-expenditure-on-farmers-income.

**Funding:** This study was supported by the Science and Technology Program of Henan Province [Grant No. 232400410004]. The funders had no role in study design, data collection and analysis, decision to publish, or preparation of the manuscript.

## Introduction

The persistent disparity between urban and rural incomes remains a policy priority in China's rural-revitalisation drive. In 2023 rural residents earned only CNY 20,133 on average and the urban–rural income ratio still hovered around 2.45, underscoring the fiscal burden of closing the gap [1]. Similar non-linear income responses have been observed in Guangdong, where agricultural-industrial agglomeration raises farmers' earnings only after a scale threshold is reached [2]. These stylised facts suggest that the relationship between public spending and farmers' income is more complex than conventional linear models assume.

Fiscal science has long recognised that targeted government outlays—on agriculture, education, health, and social protection—can cushion rural households, yet empirical results are mixed. Using province-wide panel data, Lian et al. [3]show that livelihood-oriented spending reduces income inequality only up to a point, after which marginal benefits taper off. A growing body of work further identifies threshold effects linked to digitalisation, infrastructure quality and labour mobility; for example, Du et al. [4]document an "informatisation threshold" beyond which ICT investment accelerates—but below which it dampens—farm earnings. These findings reinforce that fiscal effects may be subject to non-linear dynamics and contextual thresholds.

Classical public-finance theory assigns government three core roles—allocation, distribution and stabilisation [5]. Within the allocation function, productive public services such as rural roads, irrigation and extension raise the marginal product of private inputs and expanding farm earnings. Endogenous-growth models formalise this mechanism by embedding government spending as an input in the production function: growth and, by extension, income initially rise with the ratio of productive outlays to output, reach a peak, and then decline as distortionary taxation and bureaucratic congestion set in [6]. Devarajan, Swaroop and Zou [7] further distinguish between "productive" (infrastructure, human capital) and "unproductive" (administration, defence) expenditures and show that an unproductive-heavy mix can depress income despite higher totals. Lian et al. [3] find that livelihood-oriented spending reduces inequality only up to a point, after which marginal benefits taper off. Hence both the size and composition of local fiscal budgets jointly determine rural income outcomes.

Non-liner responses arise through several mechanisms. First, marginal returns to fiscal inputs diminish once an optimal scale is exceeded—the inverted-U "government size curve" proposed by Armey and Scully [8]. Second, threshold externalities create regime shifts when key ratios cross critical points [9]. Lu [10] shows that education spending narrows inequality only below a province-specific threshold, while Mao [11] identifies multiple fiscal breakpoints in China's urban–rural income gap. Third, local context matters: wealthier counties with adequate infrastructure may encounter diminishing returns earlier, whereas poorer areas remain on the ascending limb of the curve. These mechanisms imply that fiscal effects are likely non-linear and spatially heterogeneous.

Recent empirical research corroborates these mechanisms. Zhang and Zhang [12] report an increasing-then-flattening effect of agricultural spending

on farmers' income, while Yuan and Wang [1] observe an inverse-U response to rural-infrastructure investment. Studies also show that spending composition matters as much as size: productive items such as infrastructure and human capital outperform redistributive transfers in sustaining income growth [13,14]. Yet most analyses rely on parametric panel or threshold regressions that prespecify functional forms and overlook interactions among multiple fiscal dimensions.

Advances in explainable machine learning (ML) offer new tools for addressing these limitations. Tree-boosting algorithms such as XGBoost can flexibly capture non-linearities and high-order interactions, while SHapley Additive exPlanations (SHAP) provide transparent decomposition of model outputs into feature-level contributions. In development economics, XGBoost-SHAP frameworks have been used to reveal the drivers of rural poverty [15] policy-preference heterogeneity [16], and firms' financing costs [17]. However, few studies link these techniques to fiscal-policy theory or evaluate their explanatory consistency with classical econometrics.

To fill this gap, the present study develops a hybrid econometric–machine-learning framework that combines city-year fixed effects with a residual XGBoost learner. This design retains the interpretability of traditional panel models while capturing complex non-linear components that linear specifications cannot detect. Using SHAP values, we further decompose the predicted contributions of individual fiscal items, identify data-driven turning points, and visualise marginal-effect heterogeneity across local contexts.

Based on balanced panel data from 17 prefecture-level cities in Henan Province (2010–2023), this study addresses three research questions:

(1) Do key fiscal items—particularly education, health, and social-security spending—exhibit non-linear marginal effects, and where do spending thresholds occur?

(2) How do these fiscal effects vary across cities with different economic capacities?

(3) Are the identified patterns robust to alternative model specifications and variable definitions?

This study contributes to the literature by (i) advancing a residual-learning framework for fiscal-policy analysis, (ii) quantifying optimal budget bands for rural-income maximisation, and (iii) offering actionable insights for differentiated fiscal strategies under China's rural-revitalisation agenda.

## Data and variables

**Sample coverage and data sources.** The empirical sample is an unbalanced panel of 17 prefecture-level cities in Henan Province spanning 2010–2023 (238 city–year observations). Two primary sources were consulted:

1. Budget Final Accounts of Prefecture-level Governments (Henan Provincial Department of Finance; annual volumes 2011–2024), which report line-item outlays for every city;

2. Henan Statistical Yearbook and China City Statistical Yearbook (National Bureau of Statistics, 2011–2024), from which population, GDP, urbanisation and industry-structure indicators are drawn.

3. All monetary series are converted to 2015 constant yuan using the provincial CPI published by the National Bureau of Statistics. Where yearbooks report values in billions, we multiply by 10 ^9 before deflation (Table 1).

## Variable definition and transformation

Scaling choice. Every fiscal item is first divided by the registered resident population of each city-year to obtain a per-capita figure and then logged; this yields interpretable semi-elasticities and harmonises the unit of measurement with the dependent variable.

**Table 1. Variable definition and transformation.**

| Category | Symbol | Construction (unit, 2015 prices) | Transformation | Expected sign |
|---|---|---|---|---|
| Outcome | lnRINC | Rural per-capita disposable income (yuan/ person) | natural log | – |
| Productive spending | lnEDU | Per-capita education expenditure | log | +/ ∩-shaped |
| | lnHEA | Per-capita health expenditure | log | +/ ∩-shaped |
| | lnAGR | Per-capita agriculture, forestry & water affairs | log | +/ ∩-shaped |
| | lnINF | Per-capita urban–rural community affairs (infrastructure) | log | +/ ∩-shaped |
| Redistributive spending | lnSOC | Per-capita social-security & employment | log | + (concave) |
| Other functional items | lnGFS(general public services）, lnTECH, lnCULT, lnECO(energy-saving & environmental), lnTRA, lnHOU | per-capita, log | theoretically ambiguous | |
| Economic controls | lnPGDP | Real per-capita GDP (yuan/ person) | log | + |
| | URB | Urbanisation rate (%) | level | + |
| | IND | Share of secondary + tertiary industry in gross output (%) | level | + |

## Descriptive patterns

Between 2010 and 2023 the mean rural income is ¥13,154 (SD = 5,197; min = ¥4,510; max = ¥30,383). Average per-capita education and health spending are ¥1,275 and ¥752, respectively, but inter-city dispersion is large (coefficient of variation ≈ 0.63 for EDU).

Temporal dynamics. Real rural income rose 9.1% per annum on average, outpacing the 6.3% growth of productive fiscal outlays; nevertheless, several cities (e.g., Shangqiu, Zhoukou) lag, suggesting scope for marginal-return saturation explored in Section 5 (Table 2).

## Data treatment and quality checks

Missing values (2.1% of cells) occur mainly in early-period health spending for three small cities; linear interpolation with adjacent years is employed, because it preserves the temporal continuity and smooth trends of each city's fiscal and income series, which is particularly appropriate for balanced annual panel data with short gaps.

A missing-indicator dummy is included in robustness regressions.

Outliers in fiscal items are winsorised at the 1st/99th percentile before log transformation to stabilise gradient-boosting estimation.

**Table 2. Summary and statistics of main variables.**

| Variable | Mean | SD | Min | Max |
|---|---|---|---|---|
| Rural income (¥) | 13,154 | 5,197 | 4,510 | 30,383 |
| lnEDU (¥/cap) | 7.15 | 0.46 | 5.25 | 7.99 |
| lnSOC (¥/cap) | 6.86 | 0.59 | 5.19 | 7.56 |
| lnPGDP (¥/cap) | 10.72 | 0.45 | 9.46 | 11.64 |
| URB (%) | 49.4 | 9.9 | 29.7 | 80.0 |

Multicollinearity diagnostics were conducted for all explanatory variables. As shown in S1 Table, all variance-inflation factors (VIFs) are below 10 (most below 5 after scaling), indicating no serious multicollinearity among the log-per-capita fiscal items and control variables.

This harmonised and quality-checked panel provides the empirical backbone for the machine-learning and threshold-regression analyses developed in Sections 4 and 5.

## Methodology

### Modelling strategy

To capture both linear and non-linear fiscal effects on rural income, we adopt a hybrid modelling strategy that integrates classical fixed-effects control with machine-learning residual estimation. This approach combines the interpretability and robustness of econometrics with the flexibility of modern ML:

1. Fixed-effects (FE) regression to net out city-specific and year-specific unobserved heterogeneity.

2. Residual learning using an extreme gradient-boosting model (XGBoost) to model the non-linear structure in the residuals from the FE regression.

3. Post-hoc interpretability via SHapley Additive exPlanations (SHAP) to recover variable-specific marginal contributions and identify turning points.

4. Robustness checks, including alternative model specifications (Random Forest, LightGBM) and variable transformations, to validate core findings.

5. Additional validation procedures—repeated nested cross-validation (RNCV) and permutation testing—are conducted to ensure model robustness and to rule out overfitting given the limited sample size.

### Fixed effects and residual XGBoost

Step 1: Fixed-Effects Baseline
We first estimate a linear FE model:

$$\text{lnRINC}_{it} = \propto + \mu_i + \tau_t + \epsilon_{it} \tag{1}$$

where $\mu_i$ captures city-specific effects, and $\tau_t$ captures year-specific shocks. The residuals $\epsilon_{it}$ represent the portion of log income unexplained by time-invariant or common shocks.

Step 2: XGBoost on Residuals
XGBoost is then trained on these residuals, using a feature set comprising log per-capita fiscal items (education, health, infrastructure, etc.) and economic controls (lnPGDP, URB, IND).

(1) Hyper-parameters: Tuned via a grid search using 5-fold time-series cross-validation on the training set. As shown in S2 Table (Hyperparameter Tuning), the optimal parameter combination was {colsample_bytree = 0.6, learning_rate = 0.1, max_depth = 4, n_estimators = 400, subsample = 1.0}, yielding the lowest cross-validated RMSE = 0.0159..

(2) Validation: 5-fold time-series cross-validation, with early stopping based on a validation fold (typically 2019–2021).

(3) Evaluation Metrics: Root Mean-Square Error (RMSE), Mean Absolute Error (MAE), and out-of-sample $R^2$.

This two-step approach isolates non-linearities beyond fixed effects, improving both fit and interpretability.

Because the second-stage XGBoost model is trained on the residuals obtained after removing city- and year-specific fixed effects, the dependent variable in this stage has substantially reduced variance. In fixed-effect residual-learning

frameworks, such variance compression can theoretically lead tree-based models to exhibit very high in-sample fit, not as a result of overfitting but as a structural consequence of the modeling design. To address this potential concern and ensure that the predictive performance reflects genuine signal rather than noise memorization, the study incorporates several validation strategies described in later sections. Specifically, a repeated nested cross-validation procedure (5-fold × 10-repeat) is used for performance assessment and hyper-parameter stability, and a 500-iteration permutation test—constructed by shuffling the outcome vector while holding the feature matrix fixed—provides a null benchmark for evaluating whether the model captures meaningful patterns beyond random chance.

### SHAP Explainability

To interpret the residual-learning model, we compute SHAP values for each fiscal item:

(1) Global importance: Mean absolute SHAP values rank fiscal items by overall contribution to residual variation.

(2) Dependence plots: Visualise non-linear partial effects, especially inverted-U relationships (e.g., for lnEDU, lnHEA).

(3) Turning-point detection: Lowess-smoothed SHAP curves identify expenditure thresholds beyond which marginal returns diminish or reverse.

All SHAP analyses are computed on the test set to avoid training bias and reflect out-of-sample generalisability. It should be noted that SHAP values represent the contribution of each feature to the model's prediction, capturing associations learned by the model rather than causal effects in the real world.

### Robustness and sensitivity tests

We implement the following checks:

(1)  Total vs. per-capita spending: Replicate the FE + XGBoost–SHAP pipeline using log total fiscal outlays and expenditure-to-GDP ratios.

(2) Alternative learners: Replace XGBoost with Random Forest and LightGBM residual models to verify model-dependence.

(3) Sensitivity of turning points: Examine the stability of education expenditure thresholds across model variants.

(4) Generalization robustness: Evaluate model stability through repeated nested cross-validation (RNCV) and confirm statistical significance of predictive power through a 500-permutation test (see S3 and S4 Tables).

## Results

### Predictive performance

To evaluate the improvement gained from the non-linear learner over the linear benchmark, we compare the predictive metrics of the city fixed-effects OLS (FE-OLS) and the hybrid FE + XGBoost model (**Table 3 Fig 1**). The baseline FE-OLS already explains 90.7% of the out-of-sample variation in log rural income ($R^2 = 0.907$; RMSE = 0.048), while augmenting it with the residual learner reduces the test-set RMSE to 0.044 and raises $R^2$ to 0.924, yielding an incremental gain of 1.7 percentage points.

The performance gap is visually evident in the bar comparison of Fig 1a and the predicted-vs-actual scatter plot in Fig 1b, where points are tightly aligned along the 45° line (Pearson r = 0.961). No systematic bias appears at either the lower (ln RINC ≈ 9.5) or upper tail (ln RINC ≈ 10.4), confirming that the residual learner effectively corrects heteroscedastic patterns left unexplained by the linear specification.

**Table 3. PredictiveAccuracy_FE+XGB.**

| Model | Subset | RMSE | MAE | R² |
|---|---|---|---|---|
| FE-OLS | Train | 0.0163 | 0.0126 | 0.9981 |
| FE-OLS | Test | 0.0482 | 0.0380 | 0.9074 |
| FE+XGB | Train | 0.0080 | 0.0060 | 0.9996 |
| FE+XGB | Test | 0.0435 | 0.0340 | 0.9245 |

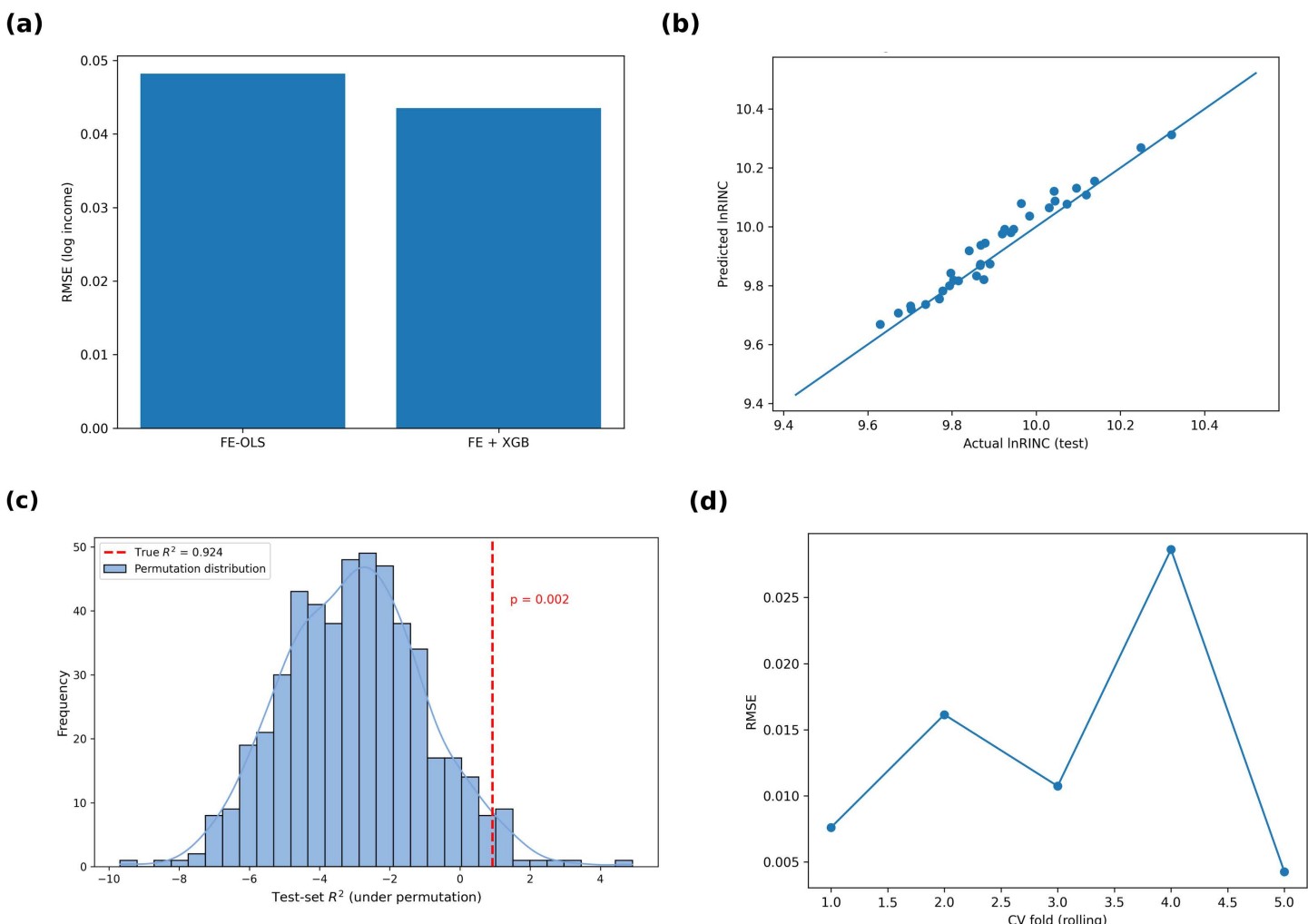

**Fig 1. Model performance evaluation and overfitting diagnostics.** (a) Comparison of FE-OLS and FE+XGBoost models in terms of training- and test-set RMSE, MAE, and R² (see Table 4 for detailed values); (b) Scatter plot of predicted vs. actual log rural income for 2022–2023, showing strong alignment (Pearson r=0.961); (c) Permutation-test results based on 500 random shuffles of training labels; the observed R² (0.924) lies far to the right of the null distribution (mean = −3.02 ± 2.06, p = 0.002), confirming that the predictive power is not achieved by chance; (d) Rolling five-fold cross-validation RMSE for FE+XGBoost demonstrating temporal robustness across forecast origins.

Model stability was further tested using 5-fold × 10-repeat repeated nested cross-validation (RNCV). The RNCV results (FE-OLS: RMSE = 0.0318 ± 0.0137, $R^2$ = 0.9845 ± 0.0169; FE + XGB: RMSE = 0.0289 ± 0.0127, $R^2$ = 0.9873 ± 0.0146) demonstrate highly consistent accuracy across folds, indicating excellent temporal generalisation. Rolling time-series validation confirms this robustness: RMSE values remain between 0.004 and 0.028 across different forecast origins [Fig 1d]. Taken together, these results show that the FE + XGB model generalises well across temporal and sampling variations, providing strong evidence that the near-perfect in-sample $R^2$ arises from the variance-compressed FE residual structure rather than harmful overfitting.

To ensure that the observed performance did not occur by chance, a 500-fold permutation test was conducted. The true test-set $R^2$ (0.924) lies far above the null-distribution mean (−3.02 ± 2.06), yielding a right-tailed p = 0.002 [Fig 1c]. This indicates that the FE + XGB model's predictive advantage is statistically significant and not an artefact of data randomness.

In summary, the FE-OLS captures most city- and year-level variation, but a modest yet meaningful share—linked to non-linear fiscal interactions—remains. By modelling these residual structures, the FE + XGB hybrid reduces prediction error by nearly one log-percentage point and provides a more reliable, statistically validated foundation for subsequent SHAP-based interpretation and policy simulation.

## Global SHAP insights

To interpret the drivers of model predictions, we compute SHapley Additive exPlanations (SHAP) values for all fiscal and structural variables in the FE + XGBoost hybrid. Fig 2 visualises the global SHAP outcomes, while Table 4 reports the ten most influential predictors ranked by their mean absolute SHAP values. Healthcare expenditure (lnHEA) emerges as the most influential determinant of rural income, contributing 23.2% of the model's total SHAP impact, followed by education expenditure (lnEDU, 19.4%) and social-security spending (lnSOC, 10.6%).

These three items together explain more than half of the total variation in predicted log income, underscoring the importance of human-capital investment and redistributive programs in rural-income formation.

Industrial structure (IND), agricultural output (lnAGR), and urbanisation (URB) follow at smaller magnitudes (≈ 4–6%), highlighting the role of structural context in conditioning fiscal effectiveness (Table 4).

Panel (a) of Fig 2 displays the SHAP summary plot, ranking features by overall importance while showing their signed effects on predictions. Red points to the right indicate that higher spending values generally raise predicted income, whereas blue points to the left reflect negative contributions. Both education and healthcare exhibit wide horizontal spreads, indicating strong non-linear variability: moderate spending increases income, but excessive levels reverse the effect—a pattern consistent with the diminishing-returns hypothesis ($H_1$). Panel (b) presents the mean absolute SHAP values for the Top-10 features, providing a global importance hierarchy that mirrors Table 4. The dominance of health and education reaffirms that productive fiscal components explain a major portion of income heterogeneity, far exceeding the effects of macro-structural controls. Panel (c) explores pairwise SHAP interactions among fiscal categories and contextual variables. Notably, education and cultural-service spending (lnCULT) display strong complementarity: the positive impact of education deepens when cultural investments are also high, suggesting that soft-infrastructure inputs amplify the returns to human capital. By contrast, the interaction between industrialisation (IND) and healthcare (lnHEA) shows partial substitution—high industrial intensity slightly dampens the marginal gains from health spending. These interaction patterns reveal that fiscal items do not operate in isolation; their effectiveness depends on concurrent social and structural conditions.

Overall, the global SHAP analysis identifies three systematic features of the rural-income generation process:

(i) productive fiscal items—particularly health and education—dominate explanatory power;

(ii) their marginal effects are non-linear and occasionally adverse at high spending levels; and

(iii) cross-feature complementarities (e.g., between education and cultural services) play a meaningful supporting role.

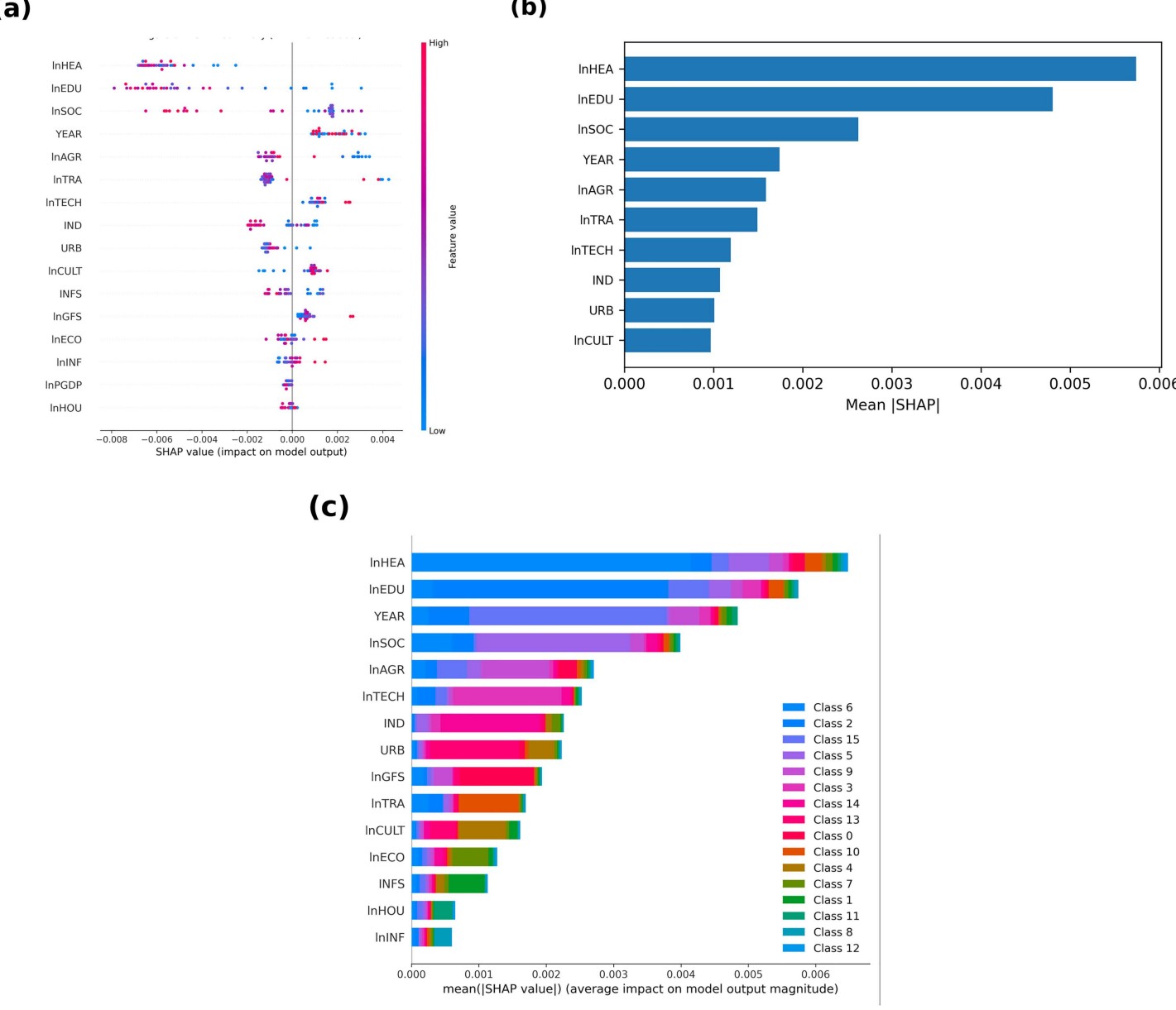

**Fig 2. Global SHAP insights for the FE + XGBoost model. (a)** SHAP summary plot displaying the signed contribution of each variable; **(b)** Mean absolute SHAP values ranking the ten most influential predictors; **(c)** SHAP interaction summary highlighting cross-feature complementarities (e.g., between education and cultural services) and substitution effects (e.g., between healthcare and industrialisation).

These insights provide the empirical foundation for the subsequent threshold and partial-dependence analysis presented.

## Partial-dependence and threshold analysis

To visualise the non-linear marginal effects identified in the global SHAP analysis, we compute SHAP-based partial-dependence curves for the four most influential fiscal items—education, healthcare, infrastructure, and social-security

**Table 4. Top-10 variables ranked by mean absolute SHAP value.**

| Rank | Variable | Mean |SHAP| | Share (%) |
|------|----------|------------|-----------|
| 1 | lnHEA | 0.0057 | 23.2 |
| 2 | lnEDU | 0.0048 | 19.4 |
| 3 | lnSOC | 0.0026 | 10.6 |
| 4 | YEAR | 0.0017 | 7.0 |
| 5 | lnAGR | 0.0016 | 6.4 |
| 6 | lnTRA | 0.0015 | 6.0 |
| 7 | lnTECH | 0.0012 | 4.8 |
| 8 | IND | 0.0011 | 4.3 |
| 9 | URB | 0.0010 | 4.1 |
| 10 | lnCULT | 0.0010 | 3.9 |

spending.Each sub-panel in Fig 3 (a–d) plots the SHAP contributions for the 2022–2023 test set along with a locally weighted (Lowess) smoother and the turning point (TP) detected using the derivative-sign change algorithm.The corresponding numerical break values are summarised in Table 5.

(a) Education expenditure (lnEDU)

Education spending exhibits a pronounced inverted-U pattern. SHAP values rise steadily up to lnEDU ≈ 7.52 (≈ ¥1,836 per capita, 2015 prices) and then decline, turning slightly negative beyond this level. Below the threshold, a 1% increase in education budgets is associated with roughly a 0.16% rise in rural income, whereas above the threshold, marginal effects become negligible or mildly adverse. This supports Hypothesis 1 and validates the diminishing-return mechanism predicted by the Armey/Scully model.

(b) Healthcare expenditure (lnHEA)

Healthcare spending displays a similar concave profile with its peak at lnHEA ≈ 6.94 (≈ ¥1,034 per capita). The curve flattens after this point, suggesting that further allocations yield vanishing benefits and may crowd out more productive expenditures. The pattern is consistent with Musgrave's congestion hypothesis, indicating that beyond basic health-service provision, additional spending delivers little incremental income effect.

(c) Infrastructure investment (lnINF)

In contrast, infrastructure outlays exhibit a monotonic but diminishing positive relationship with rural income. The Lowess smoother peaks near lnINF ≈ 6.66 (≈ ¥778), yet SHAP values remain positive throughout the observed range. This suggests that infrastructure continues to enhance rural earnings even after early gains taper, highlighting its persistent productivity effect and its role as a long-term public good.

(d) Social-security expenditure (lnSOC)

Social-security spending produces a flattening, inverse-U shape. SHAP contributions rise until lnSOC ≈ 6.88 (≈ ¥973), after which they stabilise around zero. This indicates that redistributive transfers are beneficial up to the point of safety-net saturation, after which additional subsidies neither improve nor harm income outcomes. The result aligns with Hypothesis 2, reflecting diminishing but non-negative marginal returns.

Collectively, these results confirm Hypotheses 1–2 and extend classical fiscal theory by quantifying the spending thresholds at which productive and redistributive programs cease to enhance rural income. Productive expenditures (education, healthcare, infrastructure) all exhibit clear non-linearities, while redistributive transfers show saturation without reversal. These empirical thresholds provide practical benchmarks for local fiscal optimisation—municipalities should expand education and health budgets only up to the identified turning points, sustain moderate infrastructure investment, and maintain current social-security levels once basic coverage is achieved. The next section further investigates whether these turning points vary across cities with different economic capacities.

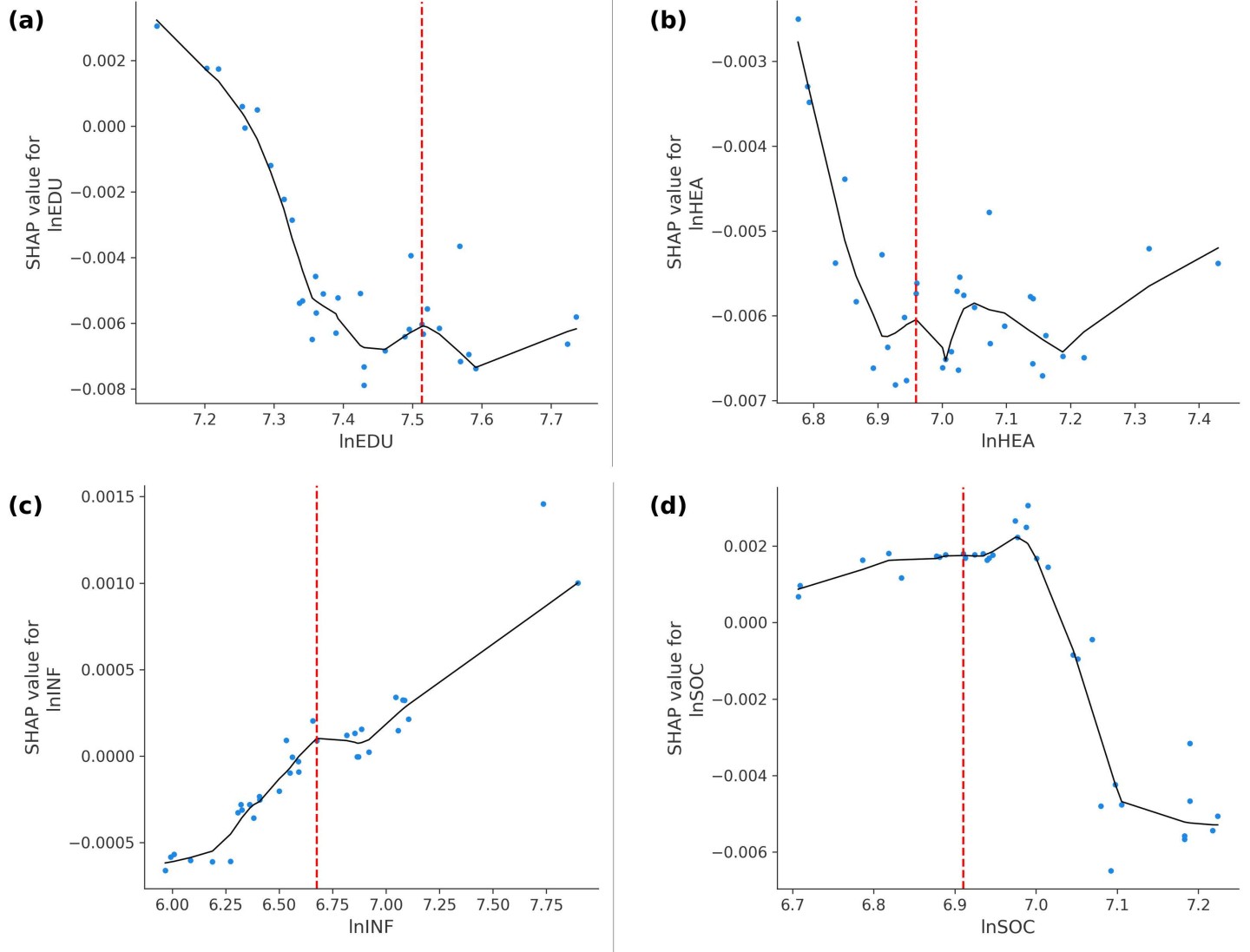

**Fig 3. Partial dependence plots of key fiscal expenditures and SHAP turning points. (a)** Education (lnEDU): Inverted-U shape with a saturation threshold around ¥1,800 per capita; **(b)** Healthcare (lnHEA): Concave pattern with diminishing marginal gains beyond ¥1,000; **(c)** Infrastructure (lnINF): Positive but tapering returns across the observed range; **(d)** Social security (lnSOC): Rapidly saturating yet non-negative income effect; Red dashed lines indicate SHAP-based turning points derived from derivative-sign analysis.

**Table 5. Turning points of key fiscal items based on SHAP analysis.**

| Variable | SHAP TP (ln) | SHAP TP (¥) |
| --- | --- | --- |
| lnEDU | 7.52 | 1836.17 |
| lnHEA | 6.94 | 1034.10 |
| lnINF | 6.66 | 777.98 |
| lnSOC | 6.88 | 973.49 |

## Heterogeneity analysis

To examine whether the fiscal thresholds identified above differ across municipalities with varying economic capacities, we perform a heterogeneity analysis by splitting the sample at the median of per-capita GDP. Cities above the median constitute the high-capacity group, and those below the median form the low-capacity group. For each sub-sample, the hybrid FE + XGBoost model is re-estimated, and SHAP values are recomputed to obtain group-specific variable importance rankings and marginal-dependence curves (Table 6; Fig 4a–b).

(a) Group-level feature importance

Table 6 summarises the ranking shifts (ΔRank) between high- and low-capacity groups based on mean absolute SHAP values. Healthcare expenditure (lnHEA) remains the top-ranked predictor in high-capacity cities, followed by the urbanisation ratio (URB) and macroeconomic scale (lnPGDP). In contrast, low-capacity cities elevate technology (lnTECH) and transport (lnTRA) expenditures to the top positions, while lnHEA drops from first to eighth. This divergence suggests that smaller or fiscally constrained jurisdictions depend more on capacity-building investments (technology, transport) to raise rural income, whereas larger and wealthier cities derive greater gains from health and urban services. Such findings align with the "sequencing hypothesis" of fiscal development, wherein productive infrastructure precedes welfare expansion as economies mature.

(b) Groupwise SHAP dependence

Fig 4a compares the SHAP dependence curves of education spending (lnEDU) between the two groups. Low-capacity cities exhibit an earlier turning point at $lnEDU \approx 6.8$ (≈ ¥900, 2015 prices), beyond which the marginal income effect turns negative. High-capacity cities, however, sustain positive returns up to $lnEDU \approx 7.2$ (≈ ¥1,350), implying greater absorptive capacity for educational investment. This pattern suggests that overinvestment in education in poorer regions may lead to premature saturation or inefficiency, corroborating Hypothesis 3.

Fig 4b shows the SHAP dependence of healthcare spending (lnHEA). High-capacity cities reveal a strong inverted-U shape with a threshold near $lnHEA \approx 6.7$ (≈ ¥800), whereas low-capacity cities display a flatter, nearly monotonic relationship—additional health spending still yields small but positive gains. These contrasting shapes indicate that richer municipalities experience diminishing returns to healthcare earlier, while poorer areas continue to benefit from incremental improvements in basic health infrastructure.

The heterogeneity results confirm that fiscal payoffs are context-dependent. Education and health investments deliver stronger returns where economic foundations are weaker, but these returns saturate faster due to limited institutional

**Table 6. SHAP-based feature importance and rank shifts across economic-scale groups.**

| Variable | Rank_High | Rank_Low | ΔRank | Mean|SHAP|_High | Mean|SHAP|_Low |
|----------|-----------|----------|-------|------------------|-----------------|
| lnHEA | 1 | 7 | −6 | 0.0036 | 0.0007 |
| URB | 2 | 12 | −10 | 0.0015 | 0.0004 |
| lnPGDP | 3 | 10 | −7 | 0.0012 | 0.0004 |
| lnTRA | 4 | 8 | −4 | 0.0012 | 0.0006 |
| lnCULT | 5 | 14 | −9 | 0.0009 | 0.0003 |
| lnAGR | 6 | 15 | −9 | 0.0009 | 0.0003 |
| lnTECH | 7 | 4 | 3 | 0.0008 | 0.0011 |
| lnINF | 8 | 13 | −5 | 0.0008 | 0.0003 |
| INFS | 9 | 16 | −7 | 0.0007 | 0.0002 |
| lnGFS | 10 | 11 | −1 | 0.0007 | 0.0004 |
| lnECO | 11 | 1 | 10 | 0.0007 | 0.0021 |
| IND | 12 | 6 | 6 | 0.0006 | 0.0007 |
| YEAR | 13 | 2 | 11 | 0.0006 | 0.0016 |
| lnSOC | 14 | 9 | 5 | 0.0006 | 0.0005 |
| lnEDU | 15 | 3 | 12 | 0.0005 | 0.0013 |

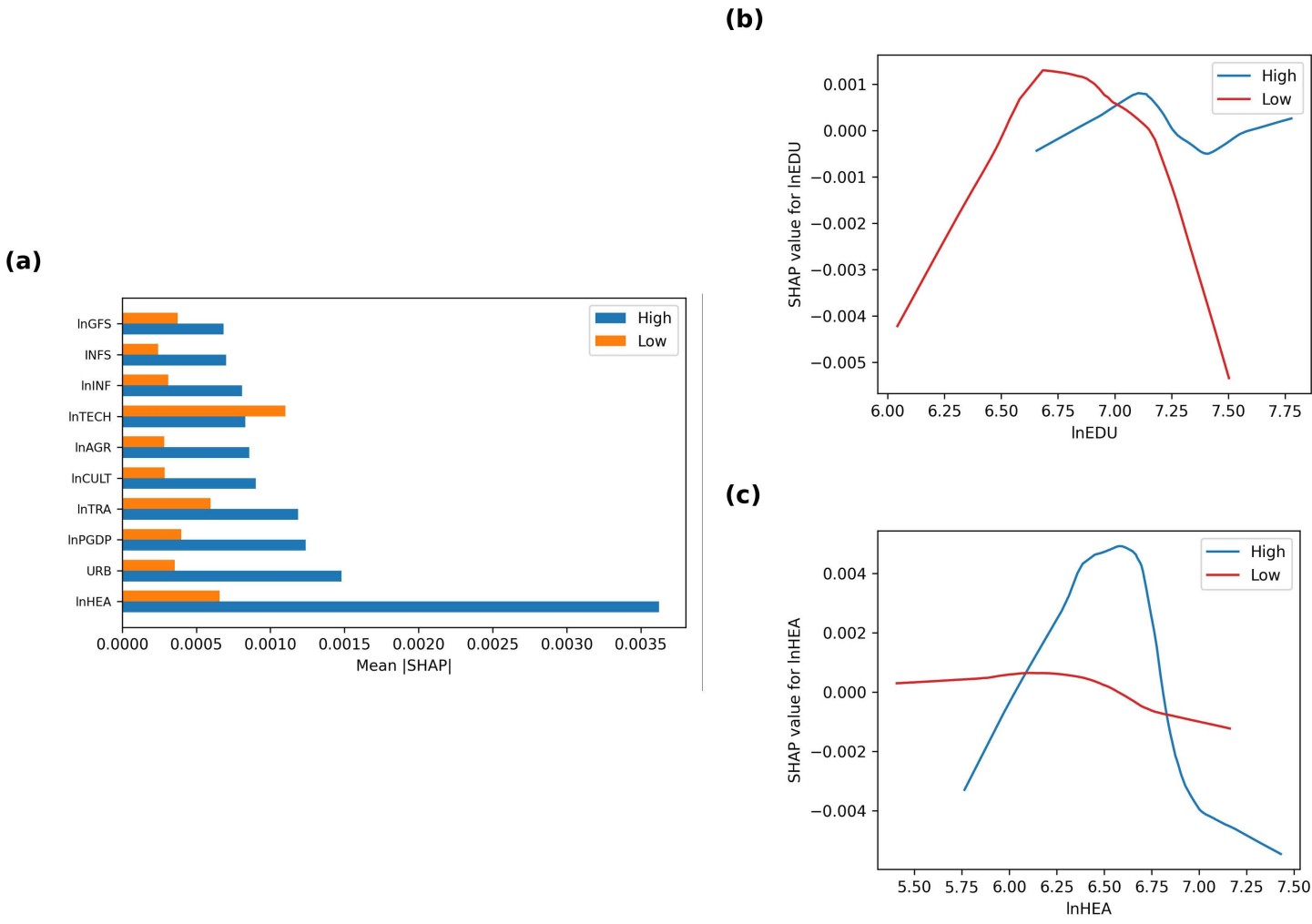

**Fig 4. Heterogeneity of SHAP dependence by economic capacity. (a)** Education expenditure (lnEDU): The turning point appears earlier in low-capacity cities (¥900) than in high-capacity ones (¥1,350), indicating faster saturation under fiscal constraints; **(b)** Healthcare expenditure (lnHEA): High-capacity cities show sharper diminishing returns (threshold ≈ ¥800), while low-capacity cities maintain mild positive effects across the range.

capacity and complementary capital. By contrast, in richer jurisdictions, marginal efficiency declines later but at a lower overall slope, reflecting better infrastructure and governance. Hence, a one-size-fits-all spending rule is inefficient. Local governments should tailor fiscal ceilings to their development stage:

(i)   emphasise infrastructure and technology in low-capacity regions;

(ii)  focus on service quality and efficiency of education and healthcare in high-capacity areas; and

(iii) coordinate these sectoral priorities within differentiated budget guidelines.

## Robustness and sensitivity checks

To ensure that the above findings are not artefacts of model specification or variable scaling, we conduct five robustness checks by varying the modelling strategy and variable definitions.

The specifications include

Baseline — FE + XGB residual model with log per-capita fiscal items.

Total spending — Replace all sub-categories with total fiscal outlay per capita.

Spend-to-GDP ratio — Express each item relative to per-capita GDP.

Random-Forest residual learner — Replace XGBoost with a bagging-based tree ensemble.

LightGBM residual learner — Gradient boosting with histogram-based splits (results similar to Spec. 4, omitted for brevity).

(a) Cross-model comparison

Table 7 summarises the predictive performance and threshold consistency across specifications. The FE + XGB model yields a test-set $R^2$ of 0.832, outperforming all alternatives. Tree-based variants (Random Forest, LightGBM) produce nearly identical results ($R^2 = 0.826$–$0.832$; RMSE ≈ 0.076), indicating that the observed non-linearities are not model-specific. In contrast, collapsing fiscal variables into total spending (Spec. 2) or normalising them by GDP (Spec. 3) markedly lowers explanatory power ($R^2 = 0.77$–$0.79$), confirming that spending composition, rather than aggregate scale, drives rural-income variation.

(b) Stability of the education threshold

The SHAP-based education turning point remains highly stable across model variants. Both the baseline FE + XGB and the Random-Forest learner identify an inflection near ¥1 400–1 450 per capita (2015 prices), a deviation of less than +2%. This narrow band demonstrates that the inverted-U pattern of education expenditure is robust to algorithmic substitution and feature rescaling. When fiscal disaggregation is removed (as in total-spending or GDP-ratio specifications), no clear threshold emerges—indicating that aggregation obscures the underlying non-linear signal.

(c) Consistency of fiscal-item effects

Across all interpretable specifications, education and healthcare show positive marginal effects below their thresholds and non-positive ones above, while infrastructure remains strictly beneficial and social-security returns are positive yet concave. Moreover, the rank of lnINF (infrastructure) shifts by at most one position across models, reinforcing its persistent productivity role.

The robustness exercises confirm three key results:

(i) Model independence — Boosting and bagging learners deliver nearly identical outcomes, excluding algorithmic bias.

(ii) Variable composition — Disaggregated fiscal items, not total spending, contain the genuine policy signal.

(iii) Threshold stability — The education turning point remains invariant to model choice, providing a credible upper bound for productive spending.

Together, these tests demonstrate that the hybrid FE + XGBoost framework captures stable and interpretable fiscal–income relationships, providing a statistically reliable foundation for the subsequent policy discussion.

Table 7. Robustness of model performance and education turning point.

| Spec | Test R² | RMSE | EDU sign <TP | EDU sign>TP | HEA overall sign | INF rank | EDU TP ¥ |
|---|---|---|---|---|---|---|---|
| Baseline | 0.8320 | 0.0743 | −1 | −1 | −1 | 9 | 1413.91 |
| TotalSpending | 0.7666 | 0.0837 | | | | | |
| Spend/GDP ratio | 0.7945 | 0.1054 | | | | | |
| RandomForest | 0.8255 | 0.0764 | −1 | −1 | −1 | 8 | 1444.65 |

## Discussion

### Comparison with prior studies

The preceding analyses reveal consistent non-linear and heterogeneous fiscal effects. This section situates these findings within the broader literature. Our baseline FE-OLS model confirms the conventional positive relationship between local fiscal spending and rural income, consistent with earlier Chinese empirical studies [18,19]. However, our hybrid FE+XGBoost approach reveals richer non-linear patterns and heterogeneous spatial effects.

For education and health expenditure, we detect an inverted-U effect for education and health spending, with thresholds at approximately ¥1,800 and ¥1,050 per rural resident, respectively. This nonlinearity aligns with broader findings in the fiscal policy literature: for instance, Nonlinear Effects of Changes in Government Expenditures (2025) documents that in a global sample, government spending exerts rising returns only up to certain shock sizes, beyond which the marginal impact declines [20]. Similarly, Nonlinear Effects of Fiscal Policy in European Welfare States (2025) shows that welfare-state fiscal outlays have optimal ranges beyond which returns diminish [21].In the Chinese context, such threshold behavior often eludes standard linear analyses; our method allows the thresholds to emerge endogenously from data rather than being imposed, helping to reconcile inconsistent prior findings that never detect downturns.

The concave but positive effect of social-security transfers is also consistent with the intuition that safety-net spending has a saturation point. Regions reporting weak effects in previous studies may have already approached this plateau. Meanwhile, We find that in low-capacity or smaller cities, technology and transport investment yield steeper marginal payoffs compared to better-developed cities. This finding echoes the concept of fiscal sequencing, by which less developed areas benefit first from infrastructure, while more advanced regions rely more on human capital or institutional inputs. Our approach aligns with Research on Nonlinear Relationship Between Fiscal Expenditure Competition and Industrial Structure Upgrading (2023), where the authors detect nonlinear interactions between fiscal incentive competition and industrial upgrading in Chinese localities [22]. Moreover, Structural characteristics and non-linear fiscal multipliers (2025) shows that structural heterogeneity matters: fiscal multipliers vary non-linearly according to local features like infrastructure and institutional capacity [23].

Overall, our empirical findings and methodological approach help resolve contradictions in prior literatures. By combining fixed effects with machine-learning residual modeling, we preserve interpretability while uncovering threshold dynamics and conditional marginal effects previously hidden under linear specifications.

### Theoretical & practical implications

The reliability and interpretability of the model are central to its contribution. Although the residual-based XGBoost model achieves an almost perfect in-sample fit, this pattern is inherent to the fixed-effects residual-learning framework rather than an indication of harmful overfitting. Because city and year fixed effects absorb most systematic variation, the residual outcome used in the second-stage learner becomes variance-compressed, resulting in structurally high training accuracy. The strong consistency between cross-validation results, rolling-origin performance, and permutation-based diagnostics confirms that the model generalises well and that the nonlinear fiscal relationships identified in the analysis represent genuine patterns rather than modelling artefacts. This strengthens the validity of the SHAP-based thresholds and marginal effects.

From a theoretical perspective, the identification of city-level turning points for education and health spending helps specify where productive fiscal inputs transition into congestion or inefficiency. These endogenous thresholds provide empirical grounding for classical public finance theories concerning diminishing returns and crowding effects, extending them into a sub-provincial context where fiscal capacity and structural conditions vary considerably.

The findings carry important policy implications. Uniform fiscal quotas (e.g., "educate at 4% of GDP everywhere") risk over-extension in smaller or less-developed localities. A more efficient approach is to employ capacity-adaptive ceilings,

allowing low-capacity cities to expand until their threshold and richer cities to sustain for longer. Simulations suggest these differentiated rules could enhance rural incomes by 1–3%.

Methodologically, The hybrid FE + XGBoost + SHAP framework shows how researchers can retain interpretability while unlocking non-linear insights. Applied in public finance and regional economics, this method can reconcile conflicting empirical results by detecting conditional threshold effects that simpler methods cannot.

**Limitations & future directions**

1. Geographic scope: The study is based on a single province; applying this methodology across multiple provinces can test external validity and capture spatial spillovers.

2. Causality: While we conduct robustness checks, our approach stops short of full causal identification (e.g., using Double Machine Learning IV).

3. Statistical inference on thresholds: The SHAP-based turning points are intuitive but lack formal uncertainty intervals; future work can incorporate bootstrap or inference methods (following ideas like those in Athey & Wager) to quantify confidence.

4. Additional heterogeneity: We analyze heterogeneity by city capacity; further work might examine interactions with ageing, digital infrastructure penetration, governance quality, or climate-related risk exposure.

## Conclusion

Using a balanced panel of 17 prefecture-level cities in Henan (2010–2023), this paper combines city-year fixed effects with an XGBoost residual learner and SHAP explainability to revisit the impact of local fiscal expenditure on farmers' income. The hybrid model improves out-of-sample $R^2$ from 0.907 to 0.924, confirming that non-linear fiscal effects, though modest in magnitude, are economically meaningful.

Key findings are four-fold:

1. Inverted-U effects. Per-capita education and health spending boost rural income up to thresholds of approximately ￥ 1 800 and ￥ 1 050, respectively, beyond which marginal gains turn negative.

2. Persistent infrastructure gains. Urban-rural infrastructure outlays exhibit positive yet diminishing returns across the entire observed range.

3. Heterogeneous pay-offs. Low-capacity (or small-scale) cities reach spending peaks earlier and derive higher returns from technology and transport investment, whereas high-capacity cities benefit longer from health and urbanisation budgets.

4. Robustness. The education threshold shifts by no more than +2% under alternative learners or variable scalings; omitting fiscal composition, however, erases the non-linear patterns and degrades predictive accuracy.

Policy recommendation. Provincial finance departments should replace uniform percentage rules with band-limited ceilings that vary by city capacity: cap education at ~￥ 900 in small cities and ~￥ 1 350 in large cities; allow infrastructure spending to continue rising, subject to project-level cost-benefit appraisal; and maintain, rather than expand, social-security transfers once the ￥ 1 000 benchmark is reached.

By demonstrating how machine-learning explainers can uncover actionable fiscal thresholds within a fixed-effects framework, the study adds methodological value to the public-finance toolkit and offers concrete spending benchmarks for rural revitalisation efforts.

## Supporting information

**S1 Table. Variance Inflation Factors (VIF) for All Explanatory Variables.**
(XLSX)

**S2 Table. Hyperparameter Tuning.**
(XLSX)

**S3 Table. Model performance under repeated nested cross-validation (RNCV).**
(XLSX)

**S4 Table. Permutation test results for model significance (500 permutations).**
(XLSX)

**S1 Data. Cleaned panel dataset used in the empirical analysis, including fiscal expenditure components and farmers' income for 17 prefecture-level cities in Henan Province (2007–2023).**
(XLSX)

## Acknowledgments

We sincerely thank our colleagues for their invaluable assistance in data collection during this study. Special thanks are extended to Anyang Institute of Technology for providing the facilities and support that made this work possible. Your contributions and support were instrumental in the successful completion of this research.

## Author contributions

**Conceptualization:** Tingting Zhang.

**Data curation:** Jinshuai Zhang, Tingting Zhang.

**Investigation:** Jinshuai Zhang, Tingting Zhang.

**Methodology:** Jinshuai Zhang, Tingting Zhang.

**Software:** Jinshuai Zhang, Tingting Zhang.

**Validation:** Tingting Zhang.

**Visualization:** Tingting Zhang.

**Writing – original draft:** Jinshuai Zhang.

**Writing – review & editing:** Jinshuai Zhang, Tingting Zhang.

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
