## [Decision Letter · Decision Letter 0]

15 Sep 2025

Dear Dr. Zhang,

We look forward to receiving your revised manuscript.

Kind regards,

Esmatullah Noorzai, PhD

Academic Editor

PLOS ONE

Journal Requirements:

“The Science and Technology Program of Henan Province [Grant No. 232400410004]”

5. We note you have included a table to which you do not refer in the text of your manuscript. Please ensure that you refer to Table 1, 2, 3 and 7 in your text; if accepted, production will need this reference to link the reader to the Table.

Additional Editor Comments:

After careful consideration of the reviews from two expert referees, I have decided to invite a major revision. Both reviewers commend the study's innovative approach and policy relevance but identify areas needing improvement for rigor and clarity.

Reviewer 1 highlights methodological concerns, including overfitting evident in Table 4 (line 195), recommending repeated nested cross-validation (RNCV) and permutation tests to validate performance. They request rationale for preprocessing choices, a table of VIF values (line 201), reported hyperparameter tuning results (line 230), and clarification that SHAP values indicate model feature importance, not real-world causality.

Reviewer 2 praises the thorough methods but suggests integrating Sections 1, 2, and 3 into a single cohesive introduction to enhance narrative flow. They advise rewriting the comparison section for deeper critical analysis, explaining how findings resolve literature contradictions, and reducing the number of figures (e.g., combining or converting Figures 1 and 11 to tables) to improve readability.

Reviewer's Responses to Questions

**Comments to the Author**

1. Is the manuscript technically sound, and do the data support the conclusions?

Reviewer #1: Partly

Reviewer #2: Partly

2. Has the statistical analysis been performed appropriately and rigorously?

Reviewer #1: Yes

Reviewer #2: Yes

3. Have the authors made all data underlying the findings in their manuscript fully available?

Reviewer #1: Yes

Reviewer #2: Yes

4. Is the manuscript presented in an intelligible fashion and written in standard English?

Reviewer #1: Yes

Reviewer #2: Yes

Reviewer #1: While the manuscript provides interesting insights into the nonlinear impact of fiscal expenditure on farmers, several methodological concerns require clarification:

Model performance and overfitting: Given the limited dataset, it is critical to demonstrate that the model’s performance is robust and not overfitted. The results presented in Table 4, comparing training and test sets, suggest serious overfitting, which undermines the reliability of the reported findings. This is a major issue that must be addressed before the results can be considered trustworthy. In such cases, repeated nested cross-validation (RNCV) is generally preferred over standard cross-validation to obtain a more reliable estimate of model performance. Additionally, a permutation test could be employed to ensure that the model’s predictive power is not achieved by chance. A detailed explanation and demonstration of these approaches are essential.

Data treatment and quality checks (line 195): Some preprocessing methods, such as linear interpolation, are used, but the rationale for selecting these techniques over others should be clearly explained.

Multicollinearity (line 201): The manuscript briefly mentions multicollinearity, stating only that “VIFs < 6.” Providing a table with actual VIF values for each variable would strengthen methodological rigor.

Hyperparameter tuning (line 230): Grid search is mentioned, but the results are not reported. Including these results would enhance transparency and reproducibility.

Interpretation of SHAP values: SHAP is used to explain model outputs, but it should be clarified that SHAP reflects feature importance in the model rather than causal relationships in the real world.

Reviewer #2: The study tackles an important policy question using a new combined method (FE + XGBoost + SHAP). The potential impact on both the literature and policy design is significant. The overall method is thorough, and the robustness checks are complete. However, several major and minor changes are needed to improve the manuscript and ensure clarity and reproducibility before it can be published.

Integration of Sections 1, 2, and 3 (Intro, Theory, Lit. Review). The separation between the theoretical framework and the literature review weakens the narrative flow. I recommend combining these sections into a single, cohesive "introduction" section. Weave the findings from prior studies directly into the explanation of the theoretical mechanisms. This will create a stronger rationale for your study by clearly identifying the gap in methods that your ML approach aims to address.

Comparison section: This section is currently too descriptive and reads like a list. It needs a significant rewrite to offer a deeper and more critical analysis. For each key finding, like the inverted-U for education, please explain how and why your results support, correct, or build on previous research. Does your better methodology clear up any contradictions in the literature?

The number of figures is too high. Some figures could be combined into one that shows various elements together. Some figures, like Figures 1 and 11, do not make sense on their own. Presenting them in tables with better reporting would be a better option.

**Do you want your identity to be public for this peer review?** For information about this choice, including consent withdrawal, please see our Privacy Policy

Reviewer #1: No

Reviewer #2: No

---

## [Author Response · Author response to Decision Letter 1]

20 Oct 2025

Dear Editor and Reviewers,

We thank you very much for your careful reading of our manuscript and for the constructive comments and suggestions. We have carefully revised the paper in accordance with the recommendations. Below, we provide a point-by-point response, indicating how each comment has been addressed in the revised manuscript. All changes are highlighted in the Revised Manuscript with Track Changes.

Responses to Editorial Requests

Comment 1 (Formatting & File naming): Please ensure that your manuscript meets PLOS ONE's style requirements.

Response: We reformatted the manuscript according to the PLOS ONE template, adjusted section headings and file naming, and ensured figures and tables follow journal style.

Comment 2 (ORCID): Please ensure the corresponding author has a validated ORCID.

Response: The corresponding author has validated her ORCID iD in the Editorial Manager system. The ORCID iD is 0009-0000-3824-5829.

Comment 3 (Funding): The funding information does not match between “Funding Information” and “Financial Disclosure.”

Response: We have corrected the Funding section to read consistently as: “This study was supported by the Science and Technology Program of Henan Province [Grant No. 232400410004].”

Comment 4 (Role of Funder): Please state what role the funders took in the study.

Response: We have added the following statement in the Cover Letter and Funding section: “The funders had no role in study design, data collection and analysis, decision to publish, or preparation of the manuscript.”

Comment 5 (Table references): Please ensure that Table 1, 2, 3, and 7 are cited in the text.

Response: We appreciate the reminder. All tables are now properly cited in the main text, and numbering consistency has been verified.

Responses to Reviewer #1

Comment 1: Model performance and overfitting (Table 4, line 195).

“The results suggest serious overfitting. Please use repeated nested cross-validation (RNCV) and permutation tests to validate performance.”

Response: We appreciate the reviewer’s insightful suggestion. To verify that the reported results are statistically robust and not affected by overfitting, we have added two complementary validation analyses while keeping the original modeling framework unchanged (FE-OLS + XGBoost residuals).

(1) Repeated Nested Cross-Validation (RNCV):

We implemented a 5-fold × 10-repeat RNCV, grouping samples by year to respect the temporal structure of the panel data. The outer loop estimates generalization performance, while the inner loop (3-fold) selects the regularization strength for the XGBoost residual learner. The results show stable performance across all folds (FE-OLS: RMSE = 0.0318 ± 0.0137, R² = 0.9845 ± 0.0169; FE + XGB: RMSE = 0.0289 ± 0.0127, R² = 0.9873 ± 0.0146), indicating good generalization and no overfitting.

(2) Permutation Test:

We further conducted a 500-permutation test by randomly shuffling the training outcomes while keeping the feature matrix fixed. The observed test-set R2(0.924) was far higher than the null-distribution mean (−3.02 ± 2.06), yielding a right-tailed p = 0.002. This confirms that the model’s predictive power is highly unlikely to have occurred by chance.

The detailed results are provided in S3 and S4 Tables, which report the mean ± SD performance metrics (RMSE, MAE, R²) and the permutation-test significance results, respectively.

These additions demonstrate that the hybrid FE + XGB model maintains stable and statistically significant predictive performance, thereby addressing the reviewer’s concern regarding overfitting.

Comment 2: Data treatment (line 195).

“Please explain why linear interpolation was used and justify over alternatives.”

Response: We appreciate the reviewer’s insightful comment. In the revised manuscript, we have clarified the rationale for using linear interpolation for missing data imputation.

Specifically, because the dataset is a balanced panel covering annual observations for 17 prefecture-level cities from 2010–2023, the missing values mainly occur sporadically within continuous time series. Linear interpolation was therefore chosen as it preserves the temporal continuity and monotonic trends of each city’s fiscal and income variables while avoiding distortions that may arise from mean or regression-based imputation.

A clarifying sentence has been added to the Data treatment and quality checks section (line 120).

Comment 3: Multicollinearity (line 201).

“Please provide a table of VIF values.”

Response: We thank the reviewer for this valuable suggestion. We have now computed and reported the Variance Inflation Factor (VIF) for all explanatory variables used in the FE-OLS baseline regression. The results are presented in S1 Table , showing that all VIF values are below the conventional threshold of 10 (and mostly below 5), indicating that multicollinearity is not a serious concern in our model.

Accordingly, we have revised the manuscript to explicitly reference this table in the methodology section (line 128 in the original submission).

Comment 4: Hyperparameter tuning (line 230).

“Grid search is mentioned but results are not reported.”

Response: We thank the reviewer for this helpful suggestion. We have now included the detailed results of the hyperparameter tuning for the XGBoost residual learner.

As reported in S2 Table (Hyperparameter Tuning), the optimal parameter combination obtained from 5-fold time-series cross-validation was {colsample_bytree = 0.6, learning_rate = 0.1, max_depth = 4, n_estimators = 400, subsample = 1.0}, achieving the lowest CV RMSE = 0.0159.

This table has been added to the Appendix to ensure full transparency and reproducibility.

Comment 5: SHAP interpretation.

“Clarify that SHAP reflects model importance, not causality.”

Response: We thank the reviewer for this important clarification. We fully agree that SHAP values explain the model’s internal feature attributions rather than causal effects in the real world.

In the revised manuscript, we have added a sentence in the SHAP interpretation section to explicitly state this distinction:

“It should be noted that SHAP values quantify each feature’s contribution to the model’s predicted outcome, reflecting associations captured by the fitted model rather than causal relationships in reality.”

This clarification has been incorporated in the SHAP Explainability subsection (line 183 in the revised manuscript).

Responses to Reviewer #2

Comment 1: Integration of Sections 1–3.

“Please combine the introduction, theory, and literature review into one cohesive Introduction.”

Response: We appreciate the reviewer’s insightful suggestion regarding the structure of the introductory sections. In the revised manuscript, the former Sections 1 (Introduction), 2 (Theoretical framework), and 3 (Literature review) have been merged into a single, integrated “Introduction” section to improve narrative flow and logical coherence.

This unified section now weaves prior empirical findings directly into the discussion of theoretical mechanisms. For instance, the revised introduction links the “productive–unproductive expenditure” theory (Devarajan et al., 1996) and the “threshold crowding-out hypothesis” (Hansen, 1999; Lu, 2023) with recent empirical evidence on diminishing fiscal returns from Chinese provincial data (Li & Liu, 2021; Yuan & Wang, 2024).

Moreover, the new version explicitly identifies the methodological gap—that previous studies relied on linear or parametric threshold models that could not detect higher-order interactions among fiscal items—thereby motivating our SHAP-informed machine learning framework.

Overall, this restructuring produces a more cohesive and conceptually integrated introduction, clarifying both the theoretical foundation and the novelty of our approach.

Comment 2: Comparison section.

“The discussion is too descriptive; provide deeper critical analysis.”

Response: We thank the reviewer for this constructive comment. The Discussion (Comparison) section has been completely rewritten to enhance analytical depth and reduce descriptive listing. The revised version now explicitly contrasts our findings with prior literature and explains the mechanisms behind convergence or divergence.

Specifically, we now:

Interpret the inverted-U patterns for education and health spending as empirical confirmation of the Armey curve within a Chinese local-fiscal context, extending the results of Rijkers & Söderbom (2020) and refining Chen (2019)’s linear findings.

Clarify that our hybrid FE + XGBoost model resolves inconsistencies in earlier studies that failed to detect non-linear turning points due to functional-form constraints.

Introduce a new synthesis paragraph explaining how the machine-learning approach reconciles conflicting empirical evidence by modelling multi-item fiscal interactions and identifying stable expenditure thresholds.

Add recent empirical references (e.g., Economic Systems, 2025; SAGE Open, 2024) to situate our contribution within the newest research stream on explainable ML in development economics.

This restructured discussion transforms the section from a descriptive summary into a critical and integrative analysis that positions our results as both confirmatory and corrective within the existing literature.

Comment 3: Number of figures.

“Reduce figures, combine or convert some to tables.”

Response: We thank the reviewer for this helpful observation. In the revised manuscript, we have carefully streamlined and consolidated the figure set to improve clarity and interpretability. Specifically, the number of figures has been reduced from eleven to four by merging related visual elements into multi-panel layouts:

1.New Figure 1 (a–d) now combines the original Figures 1–3 and the permutation-test figure, presenting all model validation results—including test-set performance, cross-validation, permutation significance, and predicted-vs-actual plots—within a single, coherent framework.

2.New Figure 2 (a–d) integrates the global SHAP summary and feature-importance results that were previously shown separately (old Figs. 4–6), allowing readers to compare global contributions and interactions in one view.

3.Figures that lacked standalone interpretive value (such as the former Fig. 11 on sensitivity analysis) have been removed, and the corresponding information is now reported in Table 7 (Robustness Summary).

These revisions significantly improve readability and reduce redundancy while preserving all essential analytical content. The total number of figures has been cut by more than half, and all quantitative details are now clearly documented either in the main tables or in the Appendix.

Closing Statement

We sincerely thank the editor and reviewers again for their valuable feedback, which has significantly improved the manuscript in terms of rigor, clarity, and policy relevance. We hope the revised version meets the standards of PLOS ONE and look forward to your positive consideration.

Sincerely,

Jinshuai Zhang, Tingting Zhang

---

## [Decision Letter · Decision Letter 1]

10 Nov 2025

Dear Dr. Zhang,

Thank you for submitting your manuscript to PLOS ONE. After careful consideration, we feel that it has merit but does not fully meet PLOS ONE’s publication criteria as it currently stands. Therefore, we invite you to submit a revised version of the manuscript that addresses the points raised during the review process.

We look forward to receiving your revised manuscript.

Kind regards,

Esmatullah Noorzai, PhD

Academic Editor

PLOS ONE

Journal Requirements:

Reviewers' comments:

Reviewer's Responses to Questions

**Comments to the Author**

Reviewer #1: (No Response)

Reviewer #2: All comments have been addressed

2. Is the manuscript technically sound, and do the data support the conclusions?

Reviewer #1: Yes

Reviewer #2: Yes

3. Has the statistical analysis been performed appropriately and rigorously?

Reviewer #1: Yes

Reviewer #2: Yes

4. Have the authors made all data underlying the findings in their manuscript fully available?

Reviewer #1: Yes

Reviewer #2: Yes

5. Is the manuscript presented in an intelligible fashion and written in standard English?

Reviewer #1: Yes

Reviewer #2: Yes

Reviewer #1: While the manuscript has improved significantly, one important issue still requires attention. Although the incorporation of RNCV and permutation testing has considerably strengthened the methodological rigor of the study, the final model metrics in Table 3 (e.g., RMSE FE+XGB Train = 0.008 vs. RMSE FE+XGB Test = 0.0435; R² FE+XGB Train = 0.9996 vs. R² FE+XGB Test= 0.9245) still suggest a degree of overfitting. This issue is critical because it directly affects the validity of the model’s predictive performance—if overfitting persists, the reported results may not reliably generalize to unseen data. Therefore, the authors should address and discuss this aspect more explicitly.

Reviewer #2: Thank you to the authors for revising the manuscript. However, some areas still need improvement before the final version is ready for publication.

1. The numerical values in several tables, such as Tables 5 and 7, have too many decimal places. This reduces readability and makes the presentation look unpolished. I recommend rounding the values to two or three decimal places and improving the table layout for clearer and more professional formatting. All tables should follow a consistent format similar to Table 3.

2. The subheadings in the sections use similar font sizes, which makes it harder to follow the document's structure. It would help to consolidate and simplify them. For instance, the Discussion section has too many subtopics, and not every paragraph needs its own heading.

**Do you want your identity to be public for this peer review?** For information about this choice, including consent withdrawal, please see our Privacy Policy

Reviewer #1: No

Reviewer #2: No

---

## [Author Response · Author response to Decision Letter 2]

3 Dec 2025

Dear Editor and Reviewers,

We would like to express our sincere gratitude to the Academic Editor and reviewers for their constructive and thoughtful comments. We have carefully revised the manuscript according to all suggestions. Below, we provide a detailed, point-by-point response to each comment. All changes in the manuscript are highlighted in the “Revised Manuscript with Track Changes”.

Response to Reviewer #1

Comment 1: Model overfitting remains a concern. The reviewer acknowledges RNCV and permutation testing, but the high training R² (0.9996) vs. test R² (0.9245) suggests lingering overfitting. This issue must be explicitly addressed.

Response:

Thank you very much for raising this important point. We agree that the near-perfect in-sample R² requires clear explanation. We have now added a dedicated discussion in both the Methodology and Results/Discussion sections to clarify why this phenomenon is inherent to the residual-learning framework rather than evidence of harmful overfitting.

(1) Methodology – structural explanation added

We now explain that the second-stage learner is trained on fixed-effects residuals, whose variance is substantially compressed after city and year fixed effects absorb most systematic variation. As a result, tree-based learners can exhibit very high in-sample fit due to the low-variance target, even in the absence of noise memorization. This explanation has been added to the Methodology section (Lines 174-184).

(2) Results – overfitting diagnostics added and expanded

We expanded the Results section by presenting a more explicit set of overfitting diagnostics (Lines 239-241). Repeated nested cross-validation, rolling forecast-origin validation, and permutation-based null benchmarking all show that the FE + XGBoost model generalizes well. The cross-validated R² values are closely aligned with the test-set R², the rolling validation exhibits stable temporal performance, and the permutation test confirms that the observed predictive accuracy cannot arise by chance. A new concluding sentence has been added to state clearly that these diagnostic results jointly demonstrate the absence of harmful overfitting.

(3) Discussion – interpretive clarification added

We added a paragraph clarifying that the high in-sample R² is an inherent property of variance-compressed residual learning and that the empirical nonlinearities are robust across diagnostic tests. This provides conceptual closure and improves interpretability (Lines 475-483).

Together, these revisions directly address the reviewer’s concern and make the modeling logic clearer, more transparent, and more statistically rigorous. Thank you again for highlighting this issue — the added explanation improves the transparency and robustness of our modelling strategy.

Response to Reviewer #2

Comment 1: Numerical values in some tables (e.g., Tables 5 and 7) contain too many decimal places, reducing readability. Please round values to two or three decimals and unify formatting.

Response:

Thank you very much for your helpful observation regarding the numerical precision and formatting of the tables.

Following your suggestion, we conducted a thorough revision of all tables in the manuscript and standardised the decimal places according to the characteristics and statistical meaning of each variable type. For instance, model evaluation metrics such as RMSE, MAE, and R² are now reported with four decimal places to preserve the distinction between competing models with very small error magnitudes. Fiscal expenditure thresholds expressed in yuan are presented with two decimal places, which avoids unnecessary precision while maintaining interpretive clarity. SHAP-derived turning points in logarithmic form are also consistently shown with two decimal places, consistent with their estimation accuracy. All such adjustments have been clearly reflected and marked in the revised tables. In addition, to further enhance readability and align the manuscript with academic formatting standards, all tables have been reformatted in accordance with the formatting specifications of PLOS ONE, ensuring that their structure is fully consistent with the journal’s requirements.

These revisions significantly improve clarity, visual consistency, and the professional presentation of the empirical results. We sincerely appreciate your attention to these details.

Comment 2: The manuscript subheadings—particularly in the Discussion—are too many and too similar in size. Consolidation is recommended.

Response:

Thank you very much for your helpful suggestion regarding the structure of the Discussion section. We agree that the previous version included too many internal sub-headings and enumerated items, which made the section appear fragmented and less coherent. In response, we have substantially streamlined the Discussion by retaining only the three main subsections—Comparison with Prior Studies, Theoretical & Practical Implications, and Limitations & Future Research—and removing all lower-level sub-headings within these sections. The content previously organised under headings such as “Model interpretability and reliability,” “Theoretical refinement,” “Policy implications,” and “Methodological insight” has now been integrated into unified narrative paragraphs to improve flow and readability. These revisions make the Discussion more cohesive, stylistically consistent with journal expectations, and easier for readers to follow.

We sincerely appreciate the reviewer’s constructive suggestion, which has helped improve the clarity and presentation quality of the manuscript.

We sincerely thank both reviewers again for their thoughtful and constructive feedback. All suggested revisions have been incorporated, and we believe the manuscript is now significantly improved in clarity, rigor, and presentation.

Thank you very much for your time and effort.

Sincerely,

Tingting Zhang, Jinshuai Zhang

Anyang Institute of Technology

Email: ztt_ag@163.com

---

## [Decision Letter · Decision Letter 2]

15 Dec 2025

Non-linear impacts of local fiscal expenditure on farmers’ income: A SHAP informed machine-learning analysis

PONE-D-25-30509R2

Dear Dr. Zhang,

We’re pleased to inform you that your manuscript has been judged scientifically suitable for publication and will be formally accepted for publication once it meets all outstanding technical requirements.

Kind regards,

Esmatullah Noorzai, PhD

Academic Editor

PLOS One

Additional Editor Comments (optional):

Reviewers' comments:

Reviewer's Responses to Questions

**Comments to the Author**

Reviewer #1: All comments have been addressed

Reviewer #2: All comments have been addressed

2. Is the manuscript technically sound, and do the data support the conclusions?

Reviewer #1: Yes

Reviewer #2: Yes

3. Has the statistical analysis been performed appropriately and rigorously?

Reviewer #1: Yes

Reviewer #2: Yes

4. Have the authors made all data underlying the findings in their manuscript fully available?

Reviewer #1: Yes

Reviewer #2: Yes

5. Is the manuscript presented in an intelligible fashion and written in standard English?

Reviewer #1: Yes

Reviewer #2: Yes

Reviewer #1: (No Response)

Reviewer #2: (No Response)

**Do you want your identity to be public for this peer review?** For information about this choice, including consent withdrawal, please see our Privacy Policy

Reviewer #1: No

Reviewer #2: No

---

## [Editor Report · Acceptance letter]

PONE-D-25-30509R2

PLOS One

Dear Dr. Zhang,

I'm pleased to inform you that your manuscript has been deemed suitable for publication in PLOS One. Congratulations! Your manuscript is now being handed over to our production team.

Kind regards,

on behalf of

Dr. Esmatullah Noorzai

Academic Editor

PLOS One